# Contrastive Learning with Adaptive Neighborhoods for Brain Age Prediction on 3D Stiffness Maps

**Jakob Träuble**                                                          *jnt27@cam.ac.uk*
*University of Cambridge*

**Lucy Hiscox**                                                      *HiscoxL@cardiff.ac.uk*
*Cardiff University Brain Research Imaging Centre*

**Curtis Johnson**                                                          *clj@udel.edu*
*University of Delaware*

**Carola-Bibiane Schönlieb**                                                *cbs31@cam.ac.uk*
*University of Cambridge*

**Gabriele Kaminski Schierle**                                              *gsk20@cam.ac.uk*
*University of Cambridge*

**Angelica Aviles-Rivero**                                                  *ai323@cam.ac.uk*
*University of Cambridge*

**Reviewed on OpenReview:** *https://openreview.net/forum?id=oI2Tpd4tiP*

## Abstract

In the field of neuroimaging, accurate brain age prediction is pivotal for uncovering the complexities of brain aging and pinpointing early indicators of neurodegenerative conditions. Recent advancements in self-supervised learning, particularly in contrastive learning, have demonstrated greater robustness when dealing with complex datasets. However, current approaches often fall short in generalizing across non-uniformly distributed data, prevalent in medical imaging scenarios. To bridge this gap, we introduce a novel contrastive loss that adapts dynamically during the training process, focusing on the localized neighborhoods of samples. Moreover, we expand beyond traditional structural features by incorporating brain stiffness—a mechanical property previously underexplored yet promising due to its sensitivity to age-related changes. This work presents the first application of self-supervised learning to brain mechanical properties, using compiled stiffness maps from various clinical studies to predict brain age. Our approach, featuring dynamic localized loss, consistently outperforms existing state-of-the-art methods, demonstrating superior performance and paving the way for new directions in brain aging research.

## 1 Introduction

Aging causes significant changes in the structure and function of the brain. Magnetic Resonance Elastography (MRE) has recently emerged as a non-invasive technique to measure mechanical brain properties (Hiscox et al., 2016), such as stiffness $\mu$, that is, the resistance of a viscoelastic material to an applied harmonic force. Studies have shown a promising age sensitivity of whole-brain stiffness measurements (Hiscox et al., 2021), surpassing established neuroimaging age biomarkers, such as volume measurements (Sack et al., 2011). With advancing MRE protocols, improvements in resolution have enabled the study of stiffness changes in more localized brain regions, revealing regional age-related variability (Murphy et al., 2013; Arani et al., 2015; Takamura et al., 2020; Hiscox et al., 2018; Delgorio et al., 2021). Furthermore, clinical studies in patients

| Method | Contrastive Regression Loss |
|--------|------------------------------|
| Rank-N-Contrast | $\mathcal{L}^{RnC} = -\sum_i \sum_{k \neq i} \log \frac{\exp(s_k)}{\sum_{x_{i,t} \in S_{i,j}} \exp(s_{i,t})}$ with $S_{i,j} := \{x_k \mid k \neq i, d(y_i, y_k) \geq d(y_i, y_j)\}$ |
| Y-Aware | $\mathcal{L}^{y-aware} = -\sum_i \sum_{k \neq i} \frac{w_{i,k}}{\sum_t w_{i,t}} \log \left( \frac{\exp(s_{i,k})}{\sum_{t \neq k} \exp(s_{i,t})} \right)$ |
| Threshold | $\mathcal{L}^{threshold} = -\sum_i \sum_{k \neq i} \frac{w_{i,k}}{\sum_t \delta_{w_{i,t} < w_{i,k}} w_{i,t}} \log \left( \frac{\exp(s_{i,k})}{\sum_{t \neq k} \delta_{w_{i,t} < w_{i,k}} \exp(s_{i,t})} \right)$ |
| Exponential | $\mathcal{L}^{exp} = -\sum_i \sum_{k \neq i} \frac{w_{i,k}}{\sum_t w_{i,t}} \log \left( \frac{\exp(s_{i,k})}{\sum_{t \neq k} \exp(s_{i,t}(1 - w_{i,t}))} \right)$ |

Table 1: Overview of Contrastive Regression Losses. This details existing methods each employing distinct strategies to refine the contrastive learning process for regression tasks.

with neurodegenerative diseases have revealed stiffness alterations exceeding those observed in healthy aging (Murphy et al., 2016; Hiscox et al., 2020a). However, current methods are limited to region-wide averages and thus fail to exploit the rich information available in stiffness maps, which can be accessed through nonlinear relationships at a voxel level.

Brain age prediction leverages neuroimaging data through machine learning by casting it as a regression problem, wherein models are trained on healthy samples to establish a baseline trajectory for aging. Regression is a statistical method used to predict a continuous outcome—such as age—based on input data, in this case, brain imaging data. This approach is particularly promising for identifying deviations from normal aging processes that might indicate neurological conditions. This approach has traditionally been applied to structural brain features using T1-weighted MRI (Baecker et al., 2021). More recently, the scope has expanded to include features obtained from resting-state fMRI, diffusion imaging and MRE are also being examined (Lund et al., 2022; Niu et al., 2020; Clements et al., 2023). Additionally, its potential in detecting and the prognosis of neurodegenerative disorders such as Alzheimer's Disease (AD) is gaining significant attention (Lee et al., 2022). In parallel, modeling approaches have evolved from traditional regression methods to state-of-the-art self-supervised learning (Zha et al., 2024; Dufumier et al., 2021; Barbano et al., 2023). Inspired by advancements in computer vision, self-supervised learning techniques, particularly contrastive learning methods, have been effectively adapted for predicting brain age from structural MRI scans (Dufumier et al., 2021; Barbano et al., 2023). Contrastive learning, a technique that learns by comparing pairs of examples, enhances model performance by structuring the embedding space to bring samples with similar ages closer together and push dissimilar samples further apart.

Despite their potential, current methods often struggle with generalization, particularly across datasets characterized by non-uniform distributions. To address this limitation, we introduce a novel contrastive loss that specifically focuses on localized sample neighborhoods. This method is distinctively designed to adapt dynamically across different stages of training, thereby enhancing model performance where traditional approaches falter. Furthermore, considering the demonstrated age sensitivity of mechanical properties compared to structural brain properties, this study is pioneering. *It represents the first application of contrastive learning to brain stiffness maps*, highlighting a new direction in neuroimaging research. Our contributions are summarized next.

● We introduce a adaptive neighborhoods approach, specifically tailored for the framework of contrastive regression learning. Our new method is designed to address the major challenge of limited generalizability in medical image analysis, particularly within datasets characterized by non-uniform distributions. By concentrating on these challenging distributions, our approach not only enhances the robustness of the models but also performance.

● We apply our method to brain stiffness maps, introducing a novel, age-sensitive modality for brain age prediction that offers insights into neurological aging processes.

● To validate our novel contrastive regression loss, we have aggregated the largest multi-study dataset of brain stiffness from healthy controls, enabling comprehensive and robust conditions. We conducted experiments and demonstrated higher performance than existing techniques.

## 2 Related Work

● **Regression Task** Regression is a statistical technique that establishes a relationship between a set of independent variables $(X)$ and a dependent variable $(Y)$, through a function $f : X \rightarrow Y$. Regression specifically addresses continuous variables, with $Y$ taking values in $\mathbb{R}$. Known for its robust effectiveness, regression has made significant impacts across various domains including (Fanelli et al., 2011; Sun et al., 2012; Lathuilière et al., 2019). Recent decades have seen a surge in research aimed at advancing deep regression techniques, significantly enhancing their performance and applicability – for example, the works of that (Gao et al., 2018; Rothe et al., 2015; Li et al., 2021; Cao et al., 2020; Yang et al., 2021). Another interesting family of techniques, which is the focus of this work, falls within the contrastive learning family. This perspective is particularly interesting due to its ability to enhance learning by emphasizing differences between samples, thereby improving model robustness and generalization (Zha et al., 2024; Barbano et al., 2023).

● **Contrastive Learning.** To construct semantically rich and structured representations, contrastive learning has become a widely adopted method for self-supervised representation learning. This technique involves differentiating between similar (positive) and dissimilar (negative) pairs of data samples $x_i$ and $x_k$, with the goal of adjusting the distances between representations in the embedding space—bringing similar items closer and pushing dissimilar ones apart (Chen et al., 2020b; Khosla et al., 2020). Contrastive learning initially gained popularity through its applications in computer vision, where it demonstrated significant improvements in tasks such as image classification and object detection. Early methods like SimCLR (Chen et al., 2020b) introduced a simple yet effective framework that maximized agreement between differently augmented views of the same data sample. This approach utilized a contrastive loss function to bring representations of augmented pairs (positive pairs) closer while pushing apart representations of different samples (negative pairs). Building on these foundations, a notable advancement was the introduction of supervised contrastive learning (Khosla et al., 2020), which extended the contrastive loss to leverage label information, treating all samples with the same label as positives. This approach enhanced the performance of models by incorporating supervised information into the self-supervised learning framework.

As we shift focus from classification to regression problems, the distinction between positive and negative pairs transitions to a continuous spectrum. This shift necessitates the model's ability to discern varying degrees of similarity, represented as $s_{i,k} = \text{sim}(f(x_i), f(x_k))$, beyond mere categorical differentiation. Particularly, we are interested in brain age prediction, which requires precise modeling of age-related changes in brain structure and function. In response to the challenge of integrating continuous labels such as age, recent advancements propose strategies such as the Y-Aware loss (Dufumier et al., 2021), which softens the boundary between positive and negative samples. Similarly, the work of that (Barbano et al., 2023) proposed the Threshold and Exponential losses, which adjust the strength of alignment and repulsion based on the similarity between continuous labels. Another work introduced the Rank-N-Contrast loss (Zha et al., 2024), which employs a comparative ranking strategy among samples. This method ranks samples based on their similarity to a given anchor, creating a ranking-based continuous spectrum of positive and negative pairs.

● **Curriculum Contrastive Learning.** Curriculum learning, which introduces training samples in a progressively challenging manner, has been effectively applied to contrastive learning to improve model generalization. In this context, simpler pairs of samples are presented early in training, with more difficult pairs introduced gradually. Chu et al. (Chu et al., 2021) proposed CuCo, a graph representation learning method that leverages curriculum contrastive learning to enhance performance on graph tasks. Similarly, Wang et al. (Wang et al., 2024) applied curriculum contrastive learning to self-supervised depth estimation under adverse weather conditions, showing improved robustness in challenging environments. These approaches demonstrate that progressively increasing difficulty in sample pairs can help models learn more effectively in both graph and visual tasks, making curriculum contrastive learning a promising strategy for various domains.

## 3 Proposed Technique

This section details the proposed adaptive neighborhoods strategy, a novel approach within contrastive learning tailored for regression tasks. The primary objective of adaptive neighborhoods is to improve the

model's generalization by progressively focusing on localized data neighborhoods in the embedding space. As training progresses, the number of repulsed samples is reduced, allowing the model to capture finer distinctions between samples, which is particularly important for handling non-uniform distributions in continuous regression tasks like brain age prediction. This approach enhances the model's ability to learn meaningful representations based on age-related similarities, regardless of specific brain regions.

## 3.1 Problem Setup

The primary challenge in brain age modeling lies in accurately mapping high-dimensional brain imaging data to a continuous age variable. Traditional contrastive learning methods are limited in their capacity to handle the subtle variations in brain stiffness associated with aging due to their global approach. Our proposed technique introduces a dynamic, localized strategy, focusing on progressively capture age-related features effectively. Importantly, our adaptive neighborhoods method progressively reduces the number of repulsed samples during training, unlike static nearest-neighbor methods like NNCLR (Dwibedi et al., 2021). This adjustment is critical for capturing the continuous nature of regression tasks, such as brain age prediction. Formally, in brain age modeling, our objective is to train a neural network capable of accurately mapping brain images $x \in \mathcal{X}$ to their corresponding target ages $y \in \mathbb{R}$, neural network capable of accurately mapping brain images, where $\mathcal{X} \subseteq \mathbb{R}^n$ represents the space of high-dimensional brain imaging data (e.g., stiffness maps). The model comprises two key components: a feature encoder $f : \mathcal{X} \to \mathcal{Z}$, which transforms brain images into an embedding space $\mathcal{Z} \subseteq \mathbb{R}^d$, and an age predictor $g : \mathcal{Z} \to \mathbb{R}$, tasked with estimating the age from these features.

## 3.2 Adaptive Neighborhoods

The adaptive neighborhoods technique is a cornerstone of our methodology, designed to optimize the contrastive learning framework specifically for the regression tasks inherent in brain age modeling. This section outlines the detailed mechanics of this technique, including model components, operational processes, and algorithmic strategies.

To enhance the precision of contrastive learning, we introduce a adaptive neighborhoods approach that progressively explores varied scales within the embedding space, as depicted in Figure 1. This methodology systematically adjusts the selection of repulsion candidates, taking into account both their proximity and the evolutionary stage of training. The selection process for repulsed samples is defined by:

$$NN(x_i; \text{epoch}) = \{x_k \mid f_{epoch}(x_k) \text{ is among the } NN_{nb}(\text{epoch}) \text{ nearest}$$
$$\text{neighbors of } f_{epoch}(x_i) \text{ based on } d(f_{epoch}(x_k), f_{epoch}(x_i))\} \quad (1)$$

This expression delineates the set of samples subject to repulsion, identified by their distance d: $\mathcal{Z} \times \mathcal{Z} \to \mathbb{R}^+$. Our approach progressively narrows the scope of nearest neighbors involved in the repulsion process, thereby focusing the learning on increasingly localized neighborhoods within the embedding space. This adaptive mechanism is governed by two critical hyperparameters: the final count of nearest neighbors, $NN_{nb,final}$, representing the ultimate scope of repulsion at the end of training, and the decrement frequency, $NN_{\text{step size}}$, which specifies the interval of epochs for adjustments in the neighbor count, as detailed in Algorithm 1.

In datasets that show non-uniform distributions, especially those with multi-modal characteristics, it is common to find some target areas oversampled and others undersampled. This scenario is typical in neuroimaging datasets (see Fig. 3). Our dynamic localized strategy is designed to address this issue. It starts by segregating distinct groups and then the training objective evolves to focus exclusively on those groups. This process is illustrated in Fig. 1. Our approach aims to reveal more coherent representations throughout the dataset.

Following the methodology proposed by (Barbano et al., 2023), we utilize kernel functions to determine the degrees of positiveness, $w_{i,k} = K(y_i - y_k)$ where $0 \leq w_{i,k} \leq 1$, between pairs of samples. This metric is defined by the function $K(y_i - y_k)$ and reflects the age similarity between two samples. A higher value of

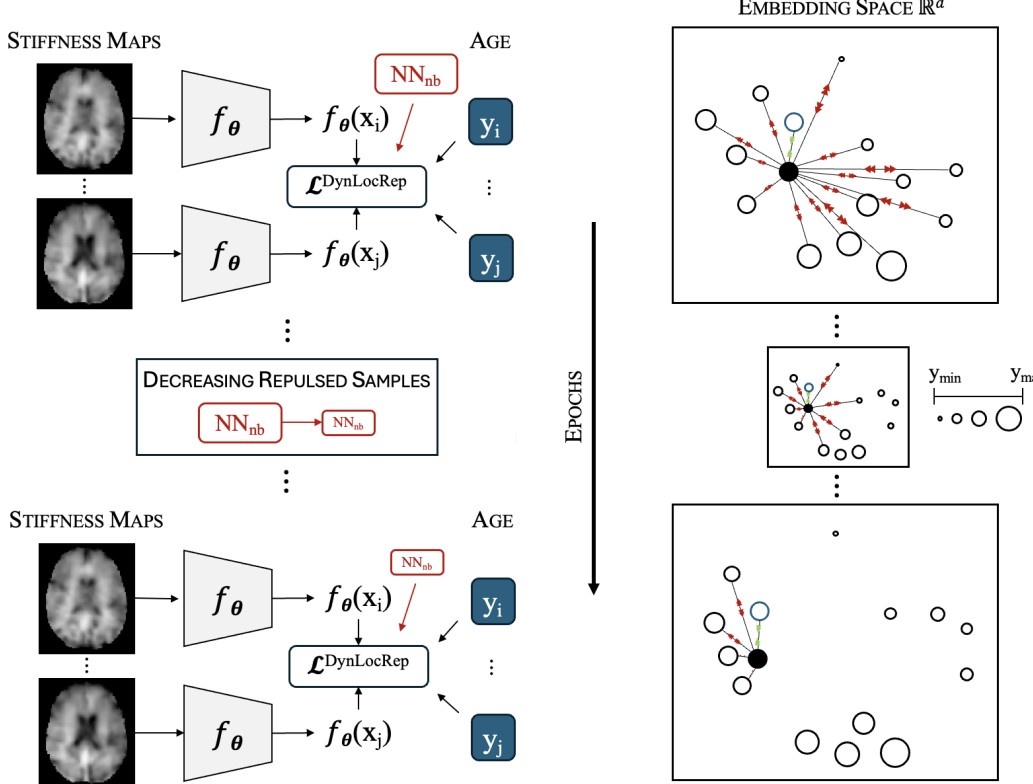

Figure 1: Graphical Illustration of Our Proposed Method. Throughout training (top to bottom), the repulsion is progressively localized as the number of samples, selected as nearest neighbors, are gradually decreased. The circle radii correspond to the label values of the samples, with larger circles representing older samples and smaller circles corresponding to younger samples.

---

**Algorithm 1** Calculation of Number of Nearest Neighbors $NN_{nb}$ (epoch)

---

**Require:** $\texttt{NN}_{\texttt{step size}} < \texttt{max epochs} \wedge \texttt{NN}_{\texttt{nb, final}} \geq 0$

$\texttt{steps completed} \leftarrow \left\lfloor \frac{\texttt{current epoch}}{\texttt{NN}_{\texttt{step size}}} \right\rfloor$

$\texttt{total steps} \leftarrow \left\lfloor \frac{\texttt{max epochs}}{\texttt{NN}_{\texttt{step size}}} \right\rfloor$

$\texttt{NN}_{\texttt{nb}} \texttt{ decrement per step} \leftarrow \frac{\texttt{batch size} - \texttt{NN}_{\texttt{nb, final}}}{\texttt{total steps} - 1}$

$\texttt{NN}_{\texttt{nb}} \leftarrow \texttt{batch size} - (\texttt{NN}_{nb} \texttt{ decrement per step} \times \texttt{steps completed})$

$\texttt{NN}_{\texttt{nb}} \leftarrow \max(\texttt{NN}_{\texttt{nb}}, \texttt{NN}_{\texttt{nb, final}})$

---

$w_{i,k}$ suggests that the samples are close in age, prompting the algorithm to minimize the distance between their representations in the embedding space. Conversely, a lower value indicates a significant age difference, thus leading to an increase in the distance between their embeddings.

Each sample $x_i$ in a batch is compared against every other sample $x_k$, with the embeddings being adjusted based on their relative age similarities. The selection of nearest neighbors for this comparison, $NN_{nb}$(epoch), dynamically changes as the training progresses. Initially, a larger set of neighbors is considered to establish broad relationships. As training advances, this number is progressively reduced to focus on more immediate and relevant interactions, thus refining the learning towards localized features. The per-sample adaptive neighborhoods loss reads:

$$l^i_{AdapN} = -\sum_k \frac{w_{i,k}}{\sum_t w_{i,t}} \log\left(\frac{\exp(s_{i,k})}{\sum_{x_t \in NN(x_i;epoch)} \exp(s_{i,t}(1 - w_{i,t}))}\right). \tag{2}$$

This expression calculates the contribution of each sample pair to the overall loss. It normalizes these contributions by the sum of positiveness weights for all comparisons within the batch, adjusting the impact of each pair based on their age-related similarity. The softmax function is then applied to these normalized and adjusted similarity scores $s_{i,k}$, defined by $s : \mathcal{Z} \times \mathcal{Z} \to \mathbb{R}^+$, which are recalculated for each dynamically defined nearest neighbor set. **What is the Intuition Behind Our Adaptive Neighborhoods?** We address the challenge of non-uniform data distributions in neuroimaging datasets. Traditional learning models often fail to adequately distinguish between different age groups when these groups are unevenly represented in the training data. Our approach refines this by adjusting embeddings dynamically. This is done via (2) that represents the per-sample loss function in a contrastive learning framework, which aims to dynamically adjust the embeddings based on age-related similarities. The essence of this equation lies in its ability to modulate the degree of repulsion or attraction between samples within the same batch based on their age proximity, which is quantified by $w_{i,k}$ the positiveness weight. *This formulation allows for adaptive learning where the focus is progressively shifted toward more challenging or informative pairs, potentially those that are not well-aligned in age, thus encouraging the model to learn finer distinctions as training progresses.*

The total loss $\mathcal{L}_{AdapN}$ is then calculated over all samples as anchors:

$$\mathcal{L}_{AdapN} = \sum_i l^i_{AdapN}$$

$$= -\sum_i \sum_{k \neq i} \frac{w_{i,k}}{\sum_t w_{i,t}} \log\left(\frac{\exp(s_{i,k})}{\sum_{x_t \in NN(x_i;epoch)} \exp(s_{i,t}(1 - w_{i,t}))}\right). \tag{3}$$

This aggregated loss function is key for structuring the embedding space optimally, ensuring that samples with similar ages are located closer together while those with significant age differences are distanced. Such a configuration not only enhances the accuracy of age prediction but also demonstrates the model's ability to learn semantically structured representations based on age-related patterns.

## 4    Experimental Results

This section details the complete set of experiments conducted to evaluate the proposed technique.

### 4.1    Data Description

We have assembled a dataset of 311 three-dimensional (3D) brain stiffness maps obtained from healthy control subjects (HC). These data were sourced from multiple clinical studies, all of which utilized highly similar Magnetic Resonance Elastography (MRE) protocols. This ensures consistency across the collected data. Although our dataset contains only 311 subjects, it represents one of the largest collections of brain stiffness maps, spanning a wide age range and offering a balanced distribution of male and female subjects. For detailed information on the individual studies and the data collection methods, please refer to Table 2. All datasets were collected in accordance with ethical standards, under protocols approved by the respective local institutional review boards.

The age distribution of all samples from each study is illustrated in Fig. 3. While age-related non-uniformity is evident in the dataset , further variability in brain coverage, as shown in Appendix Fig. 7, adds complexity to our analysis of stiffness maps. Following this, stiffness maps, depicted in Fig. 2, were processed to enhance quality and uniformity. Initially, each map underwent skull stripping using Freesurfer (Fischl, 2012), a step to isolate the brain tissue from non-relevant anatomical structures. Subsequently, we applied a bias field correction to remove intensity gradients that could affect subsequent analyses.

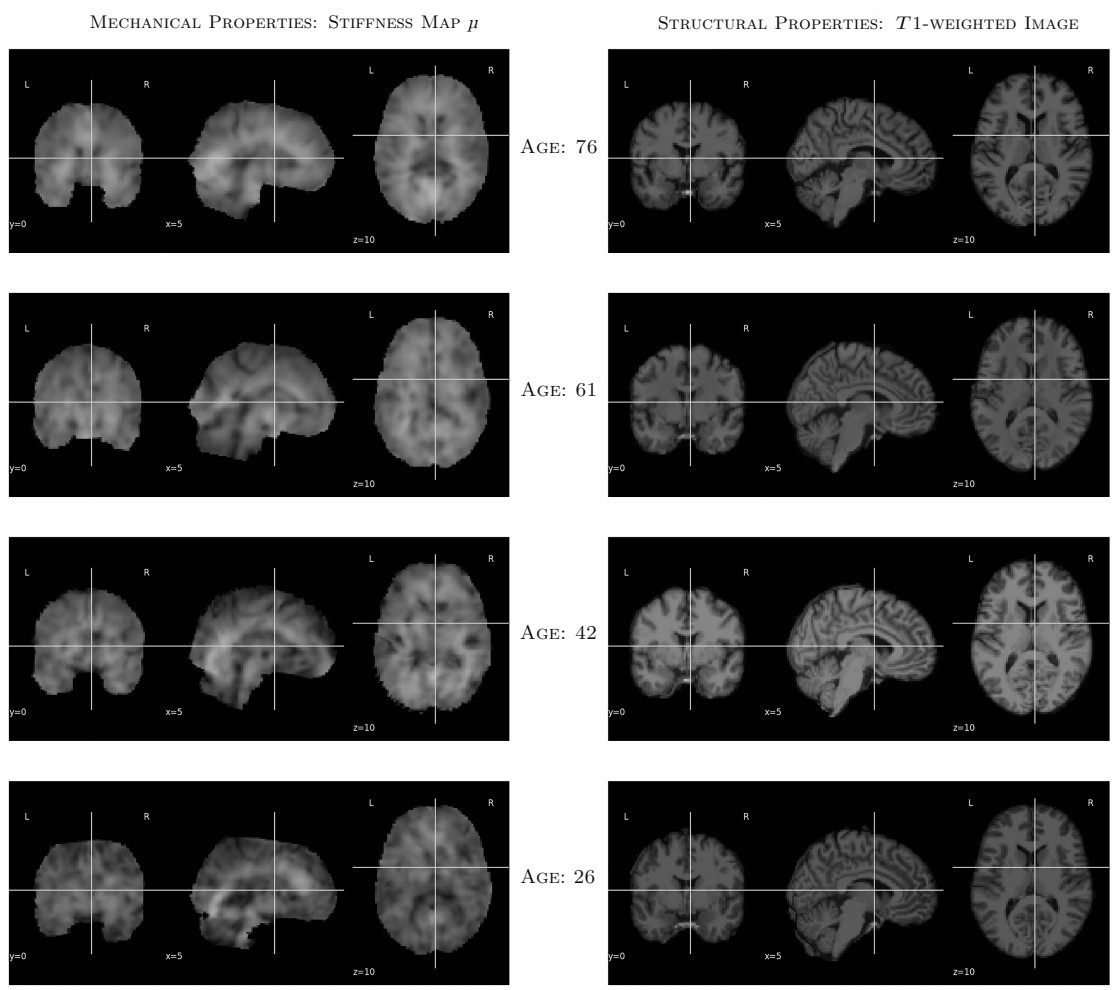

Figure 2: Comparison of Neuroimaging Modalities. Each row shows three orthogonal views (sagittal, coronal, and axial) of the brain images, highlighting the differences in mechanical and structural properties across different ages.

To address the issue of data heterogeneity across different studies, we performed affine registration of the images to the MNI152 template. This registration was conducted at an isotropic resolution of $2 \text{ mm}^3$ using ANTs (Avants et al., 2009), ensuring consistent orientation and scale among all datasets. Finally, we normalized the quantitative stiffness images, setting their mean to zero and standard deviation to one across the dataset.

## 4.2 Evaluation Protocol

Our evaluation strategy involved a 3D ResNet-18 model (33.5M parameters), which was pre-trained on the openBHB dataset (Dufumier et al., 2022), containing over 5000 $T_1$ 3D MRI brain images from multiple scanning sites, using the best reported method from the OpenBHB challenge (Dufumier et al., 2022). To enhance generalization to our stiffness dataset, we utilized quasi-raw images, ensuring uniform image pre-processing, and downsampled the structural MRI images to $2 \text{ mm}^3$ isotropic resolution. We selected ResNet-18, the smallest variant of the ResNet architecture, to match the scale of our dataset. The pre-trained ResNet-18 underwent full fine-tuning (i.e. updating all weights) on our brain stiffness dataset, following an 80:20 train-test split, over 50 epochs and a batch size of 32 using the Adam optimizer. This optimization

Table 2: Compilation of Dataset Information Across Multiple Studies: This table presents a detailed breakdown of the datasets used in our analysis. The aggregated data encompasses a diverse age range and a balanced gender ratio, facilitating a comprehensive evaluation of brain stiffness in healthy controls.

| Study | Published In | #Subjects | Age [years] | Sex [F:M] |
|---|---|---|---|---|
| 1 | (Hiscox et al., 2020c) | 134 | $23.4 \pm 4.0$ | 78:56 |
| 2 | (Bayly et al., 2021) | 60 | $37.8 \pm 20.9$ | 34:26 |
| 3 | (Hiscox et al., 2020b) | 12 | $69.4 \pm 2.4$ | 6:6 |
| 4 | (Delgorio et al., 2023) | 68 | $69.3 \pm 5.8$ | 49:19 |
| 5 | (Sanjana et al., 2021; Delgorio et al., 2022) | 37 | $49.1 \pm 16.6$ | 16:21 |
| **Total** | **-** | **311** | **41.0±21.9** | **183:128** |

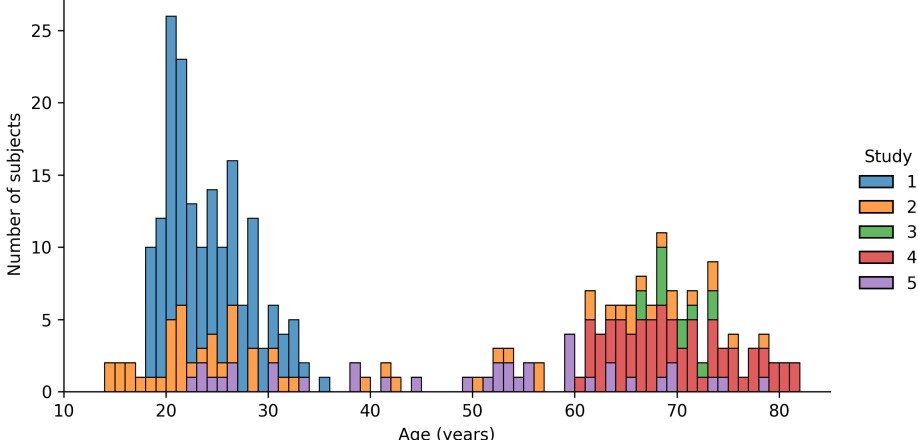

Figure 3: Age Distribution of Participants from Multi-Site MR Elastography Studies. Contribution to the 311 healthy control (HC) stiffness brain maps of different clinical studies is highlighted in color. The distribution is bimodal, indicating two predominant age groups among the subjects.

included an initial learning rate of $1 \times 10^{-4}$, decreased by 0.9 every 10 epochs, and a weight decay of $5 \times 10^{-5}$. Hyperparameters $NN_{nb,final}$ and $NN_{stepsize}$ were optimized via random search across 30 iterations. Our implementation is based on Barbano et al. (Barbano et al., 2023). As contrastive learning frameworks SimCLR and NNCLR require data augmentations, Gaussian Noise was applied for these methods. Our models were trained using an NVIDIA A100-SXM-80GB GPU. Following the training of the representations, we employed a Ridge Regression estimator (Barbano et al., 2023) on top of the frozen encoder to predict age. As an evaluation metric, we calculated the mean absolute error (MAE) on the test set, averaging the results across five random seeds.

### 4.3 Results and Discussion

We begin by evaluating the representations of stiffness maps learned using our adaptive neighborhoods loss as in (3) with Manhattan distance, $NN_{nb,final} = 14$ and $NN_{stepsize} = 1$, selected via random search, against those using current state-of-the-art contrastive classification and regression losses.

Table 3 illustrates the effectiveness of our proposed Dynamical Localized Repulsion approach in the context of brain age prediction from stiffness maps, as evidenced by the Mean Absolute Error (MAE) metric. Notably, our method significantly outperforms contrastive classification losses such as SimCLR (Chen et al.,

Table 3: Representation comparison demonstrates the superior performance of our method over state-of-the-art contrastive classification and regression losses. The best result is in green .

| Contrastive Classification Loss | MAE [years] |
| --- | --- |
| SimCLR (Chen et al., 2020a) | $9.600 \pm 1.701$ |
| NNCLR (Dwibedi et al., 2021) | $8.526 \pm 1.442$ |
| **Contrastive Regression Loss** | |
| Rank-N-Contrast (Zha et al., 2024) | $5.266 \pm 0.587$ |
| Y-Aware (Dufumier et al., 2021; Barbano et al., 2023) | $3.852 \pm 0.212$ |
| Threshold (Barbano et al., 2023) | $4.420 \pm 0.503$ |
| Exponential (Dufumier et al., 2021; Barbano et al., 2023) | $3.824 \pm 0.215$ |
| Adaptive Neighborhoods (Ours) | $3.724 \pm 0.220$ |

2020b) and NNCLR (Dwibedi et al., 2021), which achieve higher MAEs of $9.600 \pm 1.701$ and $8.526 \pm 1.442$, respectively. When comparing our method to existing state-of-the-art contrastive regression losses, our approach demonstrates superior accuracy in predicting brain age. Upon careful examination, we can observe that Rank-N-Contrast (Zha et al., 2024) shows the highest MAE, suggesting it may be less adept at capturing the nuanced patterns within the data necessary for precise age prediction. Y-Aware and Exponential (Dufumier et al., 2021; Barbano et al., 2023) losses show improvements over Rank-N-Contrast. These methods appear to better align with the underlying age-related changes in brain stiffness but still fall short compared to our approach. Threshold (Barbano et al., 2023) loss offers a competitive performance, yet it does not achieve the same level of accuracy as our method. This could indicate that while it handles some aspects of the data variability effectively, it might not fully capture the localized age-related changes as our method does. Our proposed loss achieves the lowest MAE, underscoring its ability to more accurately model the age-related changes in brain stiffness. This suggests that our method's focus on localized sample neighborhoods and its dynamic adaptation during the training process significantly contribute to its improved performance.

To investigate the role of the distance norm for nearest neighbor selection of our adaptive neighborhoods approach, we conduct an ablation study, examining the impact of various distance norms on the model's performance, as detailed in Fig. 4a. The method shows robustness regarding the choice of distance norm. Our findings reveal that the Manhattan norm achieves the lowest MAE of $3.724 \pm 0.220$ years, outperforming the Cosine (MAE = $3.748 \pm 0.142$ years), Euclidean (MAE = $3.806 \pm 0.154$ years), and Chebyshev norm (MAE = $3.842 \pm 0.196$ years).

In addition to the primary evaluation of the contrastive regression loss, we conducted a comparison of different auxiliary loss functions to investigate their impact on the final performance. Specifically, we evaluated the Mean Squared Error (MSE), Mean Absolute Error (L1), Huber loss, and DEX loss (Rothe et al., 2015). For this experiment, the encoder was first trained using $\mathcal{L}_{AdapN}$. The trained encoder was then frozen, and a separate predictor trained on top of the fixed representations using the auxiliary loss function. The results, shown in Fig. 4b, indicate that the MSE loss performs best with a MAE of $3.724 \pm 0.220$ years. Both L1 and Huber losses also showed strong performance, $3.795 \pm 0.254$ years and $3.973 \pm 0.199$ years respectively. However, the DEX loss performed significantly worse in this setup, producing a final MAE of $5.759 \pm 0.429$ years, suggesting that traditional regression-based losses like MSE are better suited for this task when combined with the contrastive learning representations compared to classification-based approaches like DEX.

Data augmentations play a prominent role in other self-supervised learning methods, as they often help models learn more robust and generalizable representations. In the context of brain imaging data, augmentation can be particularly important, but it must be applied with care due to the structural consistency of the data. To investigate this, we conducted an ablation study to explore the impact of various augmentations on the model's performance. Specifically, we applied four types of augmentations—Noise, Cutout, Rotation, and Flip—during training to examine how each one influences the learned representations and final predictions. The results, shown in Fig. 4c, indicate that Cutout provides the best performance with a MAE of

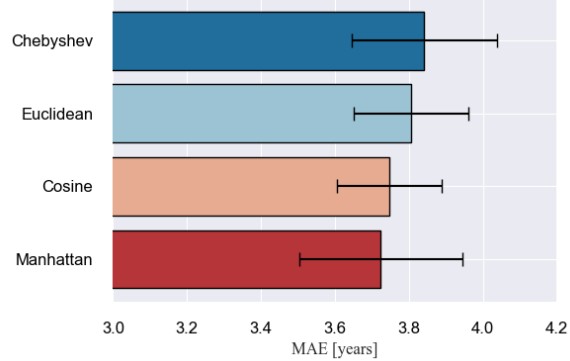 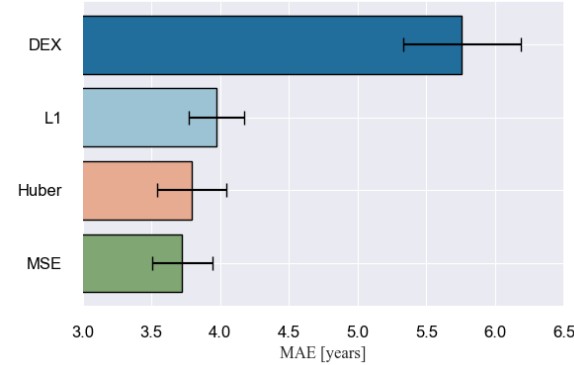

(a) Ablation study for different distance norms when selecting nearest neighbors shows Manhattan norm achieves lowest MAE.

(b) Ablation study for different regression losses shows MSE loss achieves lowest MAE compared to Huber, L1 and DEX.

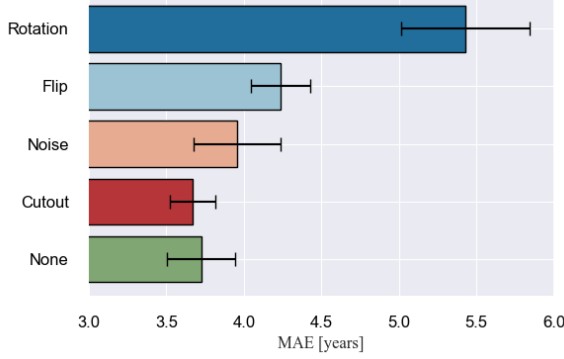 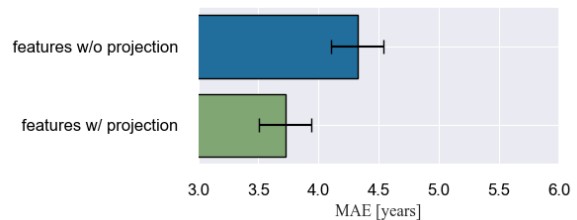

(c) Ablation study for different augmentations reveals the Cutout method achieves the lowest MAE, outperforming other augmentations.

(d) Ablation study for feature extraction shows feature extraction after non-linear projection mapping achieves lower MAE.

Figure 4: Ablation studies show the impact of distance norms, regression losses, augmentations, and projection mapping on model performance, with the Manhattan norm, MSE loss, Cutout augmentation, and projection mapping achieving the best results, respectively.

$3.667 \pm 0.147$ years. This may be due to the fact that Cutout selectively masks parts of the brain images while preserving the overall structural integrity, allowing the model to focus on the most informative areas without introducing significant distortions. Noise also performed reasonably well, with a MAE of $3.956 \pm 0.283$ years, likely because it introduces minor variations that enhance the model's robustness. Flip resulted in a slightly higher MAE of $4.238 \pm 0.192$ years. This augmentation appears to be less harmful than others, likely due to the natural symmetries in brain anatomy. In contrast, rotation had the worst performance, significantly degrading the model's accuracy with a MAE of $5.431 \pm 0.414$ years. This is likely because brain images are pre-registered during preprocessing to ensure consistent orientation across subjects, meaning that rotation disrupts this careful alignment.

In previous self-supervised learning frameworks like SimCLR, features are first extracted and then transformed through a non-linear projection before computing the loss. Specifically, the encoder $\rho : \mathcal{X} \rightarrow \mathcal{H}$ maps the input data $x \in \mathcal{X}$ to feature vectors $h \in \mathcal{H}$, where $\mathcal{H} \subseteq \mathbb{R}^d$ is the feature space. The projection function $\pi : \mathcal{H} \rightarrow \mathcal{Z}$ then maps these feature vectors to the embedding space. The overall function $f : \mathcal{X} \rightarrow \mathcal{Z}$, which maps the input $x \in \mathcal{X}$ directly to the embedding $z \in \mathcal{Z}$, is the composition of the encoder $\rho$ and the projection $\pi$: $z = f(x) = \pi(\rho(x))$. To explore the impact of this projection step in our framework, we

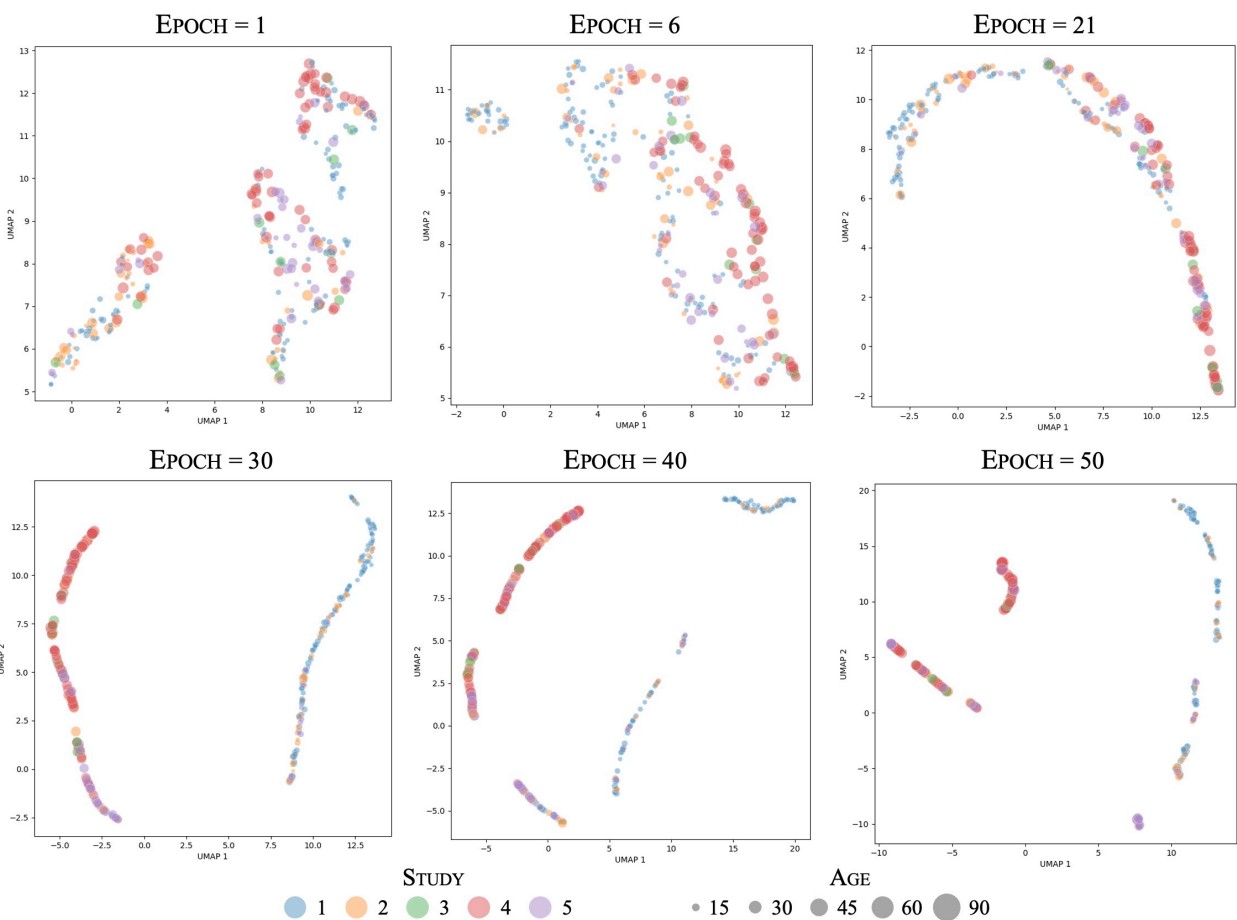

Figure 5: UMAP visualizations of representations show model improvements throughout various learning stages. As epochs increase, the clusters become more distinct and separate, indicating a more defined representation of the underlying data features.

conducted an ablation study comparing model performance under two conditions: (1) when features are used in the same space as the loss (features w/ projection), i.e. $f(x)$ is equivalent to $\rho(x)$, meaning the embeddings directly correspond to the features, and (2) when features are computed before the projection step (features w/o projection), making the embeddings distinct from the features. The results, shown in Fig. 4d, indicate that the model benefits from extracting the features after the projection layer, achieving a MAE of $3.724 \pm 0.220$ years with the projection, compared to $4.323 \pm 0.216$ years.

Following the ablation studies, we turn our attention to the evolution of the representations as the model learns across epochs. The UMAP embedding visualizations presented in Fig. 5 showcase the progressive refinement of the feature space through epochs 1, 6, 21, 30, 40 and 50. Initially, the representations are scattered and lack clear structure, indicating that the model is yet to learn distinct age-related patterns. As training progresses, we observe the emergence of more defined clusters, reflecting the model's increasing adeptness at capturing age-related variations in brain stiffness. By epoch 50, the embeddings show distinct, well-separated groupings, suggesting that our method has achieved a better understanding of the underlying age-related features. These visualizations not only confirm the effectiveness of our adaptive neighborhoods approach but also offer intuitive insights into how contrastive learning can be harnessed for regression tasks in medical imaging.

## 5 Conclusion

We introduced a novel contrastive regression loss that adeptly prioritizes local regions within embedding spaces and dynamically adjusts these regions throughout the training process. By applying this method to brain stiffness maps obtained from Magnetic Resonance Elastography (MRE), we achieved superior predictive performance in brain age estimation compared to established contrastive learning methods. This advancement not only demonstrates our model's efficacy but also underscores the potential of using localized dynamic adjustments in the analysis of complex neuroimaging data. Significantly, *our research marks the first application of self-supervised learning techniques to explore the mechanical properties of the brain, an area previously uncharted in the literature.* The implications of this are profound, opening up new avenues for understanding the structural changes associated with aging and potentially other neurological conditions. However, while our proposed contrastive learning method shows strong predictive performance in brain age estimation, its validation remains limited to the domain of brain imaging. Broader testing across diverse datasets and comparison with other approaches, such as purely supervised learning, are needed to fully substantiate its generalizability. Thus, while promising, our method requires further exploration to confirm its broader applicability. Future work includes to extend our framework to include cohorts with neurological diseases, such as Alzheimer's and Parkinson's. This expansion is expected to provide deeper insights into the progression and early diagnosis of these conditions, leveraging the detection capabilities of our model. Additionally, we aim to explore the integration of multimodal imaging data to enhance the robustness and accuracy of our predictions, potentially leading to breakthroughs in personalized medicine and neuroimaging analytics. Finally, as this is a first strategy to leverage non-nearest samples, we acknowledge that more advanced methods to introduce bias in the sampling strategy could be explored. This direction, including adaptive or more dynamic reweighting mechanisms, is an exciting area for future work.

## Acknowledgments

JT acknowledges support from the Gates Cambridge Trust via the Gates Cambridge Scholarship. CJ acknowledges partial support from the National Institutes of Health grants R01-AG058853 and U01-NS112120. CBS acknowledges support from the Philip Leverhulme Prize, the Royal Society Wolfson Fellowship, the EPSRC advanced career fellowship EP/V029428/1, EPSRC grants EP/S026045/1 and EP/T003553/1, EP/N014588/1, EP/T017961/1, the Wellcome Innovator Awards 215733/Z/19/Z and 221633/Z/20/Z, CCMI and the Alan Turing Institute. GSKS acknowledges funding from the Wellcome Trust (065807/Z/01/Z) (203249/Z/16/Z), the UK Medical Research Council (MRC) (MR/K02292X/1), ARUK (ARUK-PG013-14), Michael J Fox Foundation (16238; 022159), and Infinitus China Ltd. LVH is supported by the Wellcome Trust (grant number: 226420/Z/22/Z). AAR gratefully acknowledges funding from the Cambridge Centre for Data-Driven Discovery and Accelerate Programme for Scientific Discovery, made possible by a donation from Schmidt Futures, ESPRC Digital Core Capability Award, and CMIH and CCIMI, University of Cambridge.

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

# A    Appendix

## A.1    Influence of Sampling of Non-Nearest Neighbors

To study the effect of including farther examples at reduced rates, we introduce a distance-based weighting scheme for the non-nearest samples. The weight $\nu_{i,t}$ assigned to each non-nearest neighbor is determined by the function:

$$\nu_{i,t}(d_{i,t}) = (\frac{1}{d_{i,t}})^\lambda \tag{4}$$

where $\frac{1}{d}$ represents the normalized inverse distance between the anchor and the sample, and $\lambda$ is a hyperparameter controlling the rate of decay in weighting for samples at greater distances. Thus, the repulsion term in 3 is expanded to all samples, and non-nearest samples accordingly weighted. This formulation ensures that as $\lambda$ increases, the contributions from farther negative samples diminish more rapidly. In the extreme case where $\lambda \to \infty$, only the nearest neighbors contribute to the loss 3. Conversely, when $\lambda = 0$, the weighting becomes uniform across all samples, recovering the exponential contrastive regression loss (Dufumier et al., 2021; Barbano et al., 2023). We performed an ablation study to investigate the impact of different values of $\lambda$ on the model's performance, as illustrated in Fig. 6. The results indicate that while re-weighting the non-nearest samples provides some benefits, only one configuration with an intermediate value of $\lambda = 1$ marginally outperformed the approach of simply dropping non-nearest samples yielding a MAE of $3.681 \pm 0.234$ years. This suggests that re-weighting can help retain informative negative samples but has a limited overall impact on improving model generalization. Further exploration of more sophisticated re-weighting mechanisms could be beneficial in future work.

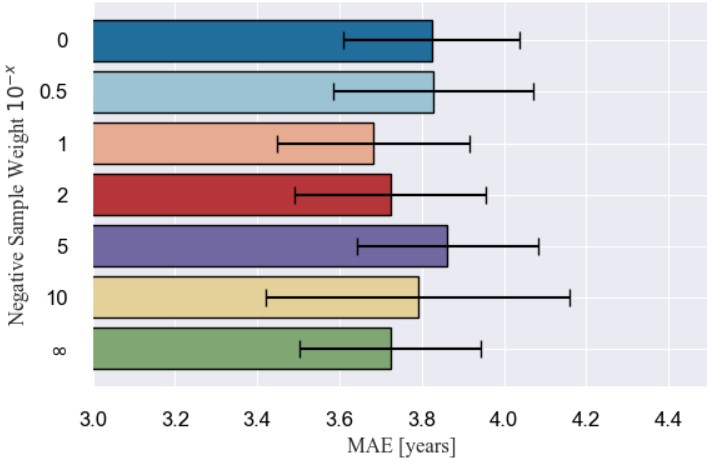

Figure 6: Ablation study for sampling of non-nearest neighbors.

## A.2 Brain Coverage of Stiffness Maps

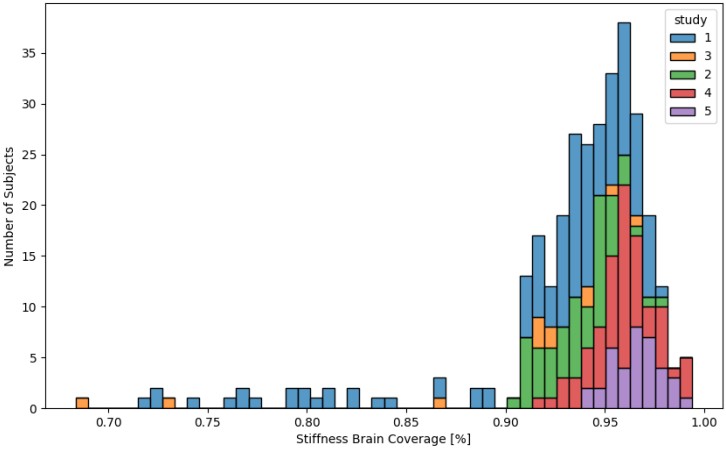

Figure 7: Distribution of Brain Coverage in Stiffness Maps Across Studies. In Magnetic Resonance Elastography (MRE), achieving optimal brain coverage necessitates a balance between high spatial resolution, signal-to-noise ratio (SNR), and acceptable scan times (Johnson et al., 2014). The majority of existing brain MRE studies focus on deep brain structures, leading to limited coverage in certain areas. This limitation is illustrated here, where the brain coverage variability in our stiffness maps is visualized.

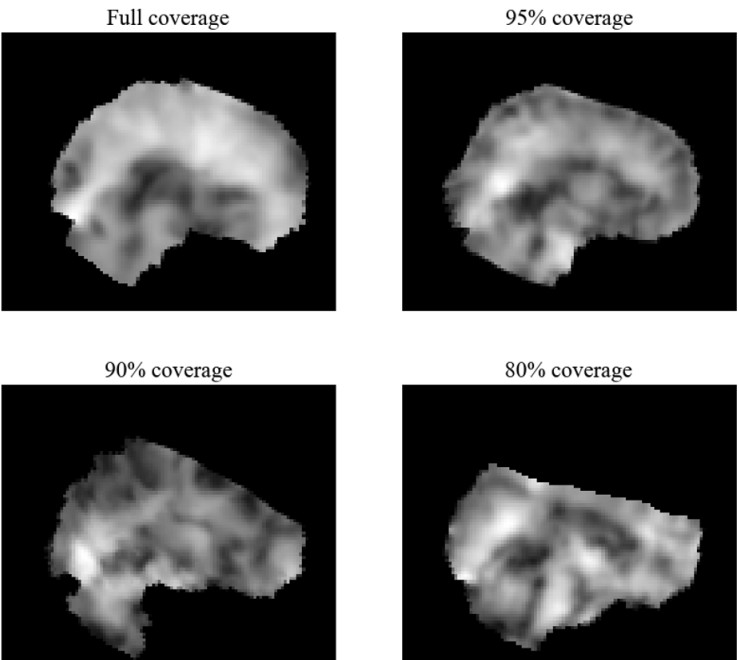

Figure 8: Examples of stiffness maps across a spectrum of coverage percentages. Full coverage is shown alongside three reduced coverage cases: 95%, 90%, and 80%, respectively. As seen in Fig. 7, most of the data fall between 90% and full brain coverage.

