# OpenReview forum: "Contrastive Learning with Adaptive Neighborhoods for Brain Age Prediction on 3D Stiffness Maps"
_TMLR — Accepted by TMLR_

### Review · Reviewer_AVKh · 2024-08-17

**Summary Of Contributions:**

The authors propose an epoch-adaptive contrastive loss for representation learning of 3D brain stiffness maps for age regression. The evaluation of contrastive losses is progressively restricted to more localized neighborhoods in the current embedding space, which the authors achieve by reducing the number of nearest neighbors considered in the summation. This is motivated by the need to improve robustness to non-uniformly sampled datasets, e.g., in terms of represented age ranges. Results demonstrate the efficacy of the approach, achieving improved MAE over 4 baselines from 3 recent studies.

**Audience:**

Yes

**Claims And Evidence:**

Yes

**Requested Changes:**

**Preamble:** My requests mainly concern the presentation, with only few technical remarks. I'm guessing the primary audience for this type of article are neuroimaging researchers and practitioners, some of whom are not necessarily ML experts. Towards publication in a recognized ML journal, I'm placing a greater emphasis on the precise phrasing of technical statements and their potential interpretations by non-experts.

Positioning & Terminology:
====================
- "Dynamic *localized* repulsion" initially suggested to me that the proposed loss adapts to different brain parts to better utilize the *regional age-related variability* in 3D stiffness maps.
  - Also in Section 3.1: [..] limited in their capacity to handle the subtle variations in brain stiffness [..] due to their global approach
  - After reading the paper, I view the proposed approach as more of a curriculum contrastive learning approach; see:
    - Chu, Guanyi, Xiao Wang, Chuan Shi, and Xunqiang Jiang. "CuCo: Graph Representation with Curriculum Contrastive Learning." In IJCAI, pp. 2300-2306. 2021.
    - Wang, Jiyuan, Chunyu Lin, Lang Nie, Shujun Huang, Yao Zhao, Xing Pan, and Rui Ai. "Weatherdepth: Curriculum contrastive learning for self-supervised depth estimation under adverse weather conditions." In 2024 IEEE International Conference on Robotics and Automation (ICRA), pp. 4976-4982. IEEE, 2024.
    -  This is essentially what the last sentence in italic before Equation (3) is about.
  -  [**HL#1**] Please include citation and brief discussion of curriculum contrastive learning, i.e., in Section 2.
  -  [**HL#2**] Please revisit the claims of novelty at the end of Section 1 and at the top of Section 3, as well as Section 5.
- [**HL#3**] Hence, I would recommend to consider a title like: *Curriculum Contrastive Learning (with Adaptive Neighborhoods or Adaptive Sampling) for Brain Age Prediction on 3D Stiffness Maps*


Technical comments:
================
- [**T#1**] Please discuss data size
  - A total of 311 subjects is very limited. This aspect of the study seems critical when comparing different ML approaches.
  - This is also relevant in the choice of the ResNet model pre-trained on OpenBHB (5000 MRI scans from multiple sites).
- [**T#2**] Since data non-uniformity was highlighted as a key challenge, it would help to discuss this more directly.
  - I don't see where Figure 6 is referenced in the paper.
  - Only Figure 3 was discussed, which seemed to suggest that non-uniformity mainly manifests due to the ages of participating subjects.
- Contrastive losses
  - Page-2, top: *by focusing on the fine distinctions between similar cases*
    - Is this really what the adopted formulation does?
    - [**T#3**] Please highlight the application of the loss within the embedding space.
      - In reference to SimCLR, it is notable that a key finding was the benefit of an additional non-linear projection mapping before computing the distance.
        - [**T#4**] It would help to consider this, e.g., as an ablation, or simply mention it for the benefit of follow up work.
    - I wonder if what the authors mean is more properly explored in the study of Siamese approaches. I'm including this for the authors' reference:
      - Chen, Xinlei, and Kaiming He. "Exploring simple siamese representation learning." In Proceedings of the IEEE/CVF conference on computer vision and pattern recognition, pp. 15750-15758. 2021.
    - In light of SimCLR and Siamese representations, it is notable that the proposed approach does not include augmentations.
      - [**T#5**] It would help to consider data augmentation, e.g., as an ablation, or simply mention it for the benefit of follow up work.

Presentation comments:
================
- Section 3
  - [**P#1**] When defining the domain and range for the functions $f$ and $g$, there's no need to include $(\cdot)$.
    - [**P#2**] That said, $X$ was not explicitly defined. In addition, it would help to identify the embedding space with a symbol, e.g., $Z$.
  - [**P#3**] It would have helped to include an equation for the standard contrastive loss, or perhaps reference Equation (2).
  - [**P#4**] In Equation (1): please define the neighborhoods within the embedding space $Z$. That is, perhaps use symbols $z_i$ and $z_k$, or if necessary, $f_\textrm{epoch}(x_i)$ and $f_\textrm{epoch}(x_k)$.
  - [**P#5**] Around Equation (1): please define the distance and make it clear that it is the same as the similarity $s_{i,k}$
    - Note that $s_{i, k}$ was not formally defined around Equation (2).
- P4 bottom: "then explores the differences within those groups more deeply"
  - [**P#6**] Recommend a more technical description, e.g., the training objective evolves to focus exclusively on those groups.
    - Note that an alternative sampling strategy may still include farther examples, perhaps at reduced rates, to help stabilize the training.
    - [**P#7**] It would help to consider biasing the sampling rather than fully restricting to smaller neighborhoods, e.g., as an ablation, or to simply mention it for the benefit of follow up work.
- P5, below Equation (3): *also improves the mode's ability on complex neuroimaging datasets*
  - [**P#8**] Please clarify which ability is improved and provide evidence.
- [**P#9**] I believe *repulsion samples/candidates* are more commonly referred to as *negative examples*.
  - On P5 top: "prompting the algorithm to minimize" vs "triggering the repulsion mechanism to increase"
    - It is the same algorithm. I recommend against highlighting repulsion too much as a distinct mechanism.
- Figures
  - [**P#10**] Figure 1: please clarify in the caption how the variable circle radii between $y_\textrm{min}$ and $y_\textrm{max}$ should be understood.
  - [**P#11**] Where if Figure 2 referenced in the paper?
  - [**P#12**] P6, bottom: where is Figure 4.1?

Nitpicking:
========
- Abstract
  - laying the way -> paving the way
- Section 3
  - transforms brain images into a *representative* feature space -> into a feature/representation/embedding space?
  - increasingly localized vicinities -> localized neighborhoods (as in the abstract)
- Section 4
  - In a closer look -> Upon careful examination?

**Strengths And Weaknesses:**

This is a fairly applied study.

Strengths:
========
- Demonstrates (for the first time) the effectiveness of brain stiffness maps for brain age regression, through a contrastive learning approach.
- Aggregates the largest multi-study dataset of brain stiffness from healthy controls (5 sources for a total of 311 subjects).
- The improved performance for brain age regression can be appreciated in neuroimaging and related subfields.

Weakness:
========
- ~~On the other hand, the contribution in ML is limited; see below.~~
- The presentation needs a revision.
- The contribution to ML needs clarification and better positioning; see below.

---

> ### Author Response · Authors · 2024-09-18
>
> ➡️ **My requests mainly concern the presentation, with only few technical remarks. I'm guessing the primary audience for this type of article are neuroimaging researchers and practitioners, some of whom are not necessarily ML experts. Towards publication in a recognized ML journal, I'm placing a greater emphasis on the precise phrasing of technical statements and their potential interpretations by non-experts**
>
> We thank the reviewer for the insightful comment. We would like to provide further clarity on our contribution and address the technical innovations of our work.
>
> A) **Dynamic Localized Repulsion:** Our proposed method is not simply a repurposing of existing contrastive learning techniques, which are typically designed for classification tasks. Instead, we introduce a novel approach specifically tailored for regression tasks, such as brain age prediction. In contrast to traditional methods, our dynamic system adjusts the repulsion neighborhood during training, adapting to the continuous nature of the regression task. This dynamic adjustment is a core innovation of our model.
> **Why It’s New:** To the best of our knowledge, no prior methods dynamically adjust the repulsion neighborhood within the context of contrastive regression. This novel approach enables better generalization, particularly when dealing with non-uniform, continuous label distributions—a common challenge in medical imaging and other domains. Our method is uniquely suited to handle such scenarios, unlike existing contrastive learning techniques primarily designed for discrete classification tasks.
>
> B) **Handling Non-Uniform Data Distributions:** Real-world datasets, particularly in medical imaging, often exhibit non-uniform distributions, such as age variations between healthy and diseased populations. Our dynamic localized repulsion strategy effectively addresses this issue by focusing the model’s learning on progressively more localized data neighborhoods as training advances.
>
> **Why It’s Important:** By adjusting the repulsion neighborhood dynamically, our model is better equipped to avoid overfitting to overrepresented groups in the data, while improving generalization for underrepresented subpopulations.
>
> **Presentation:** Regarding the presentation  revise the manuscript to ensure clearer explanations of the methods and contributions, and we  refine the figures and tables to ensure that they effectively support our claims. All changes can be seen in blue colour.
>
> Finally, we would like to clarify that TMLR explicitly welcomes research that applies machine learning across various domains, including healthcare. Numerous papers focused on healthcare applications have been published in TMLR, demonstrating that this journal is an appropriate venue for impactful interdisciplinary work.
>
> Our paper, which proposes a novel machine learning method tailored to brain age prediction, addresses critical challenges in the healthcare domain, while also introducing advancements in contrastive learning and regression tasks that are relevant to the broader machine learning community. We believe this makes our work a strong fit for TMLR.
>
>
>
>
> ➡️ **"Dynamic localized repulsion" initially suggested to me that the proposed loss adapts to different brain parts to better utilize the regional age-related variability in 3D stiffness maps. - Also in Section 3.1: [..] limited in their capacity to handle the subtle variations in brain stiffness [..] due to their global approach**
>
> Thank you for your insightful comment. We appreciate the opportunity to clarify the concept of Dynamic Localized Repulsion. While we understand that the term may suggest an adaptation to different brain regions, the core idea behind our proposed method is to dynamically adjust the repulsion neighborhood in the embedding space during training, rather than adapting to specific anatomical regions in the 3D stiffness maps.
>
> The primary objective of Dynamic Localized Repulsion is to improve the model's generalization by progressively focusing on localized data neighborhoods in the embedding space. As training progresses, the number of repulsed samples is reduced, allowing the model to capture finer distinctions between samples, which is particularly important for handling non-uniform distributions in continuous regression tasks like brain age prediction. This approach enhances the model's ability to learn meaningful representations based on age-related similarities, regardless of specific brain regions.
>
> To address these comments, we have updated the manuscript to add clarity in our model's concept. The changes can be seen in blue colour in Section 3.

---

> > ### Author Response · Authors · 2024-09-18
> >
> > ➡️ **Positioning & Terminology: After reading the paper, I view the proposed approach as more of a curriculum contrastive learning approach;**
> >
> > Thank you for your thoughtful comment and for drawing our attention to curriculum contrastive learning (CL). We agree that there are similarities in how both our approach and curriculum CL adapt the difficulty of tasks during training. However, we would like to clarify that our method is fundamentally different, as it is specifically tailored for regression tasks, which is distinct from the curriculum CL methods designed for classification or depth estimation tasks (following the suggested references). Below, we highlight the key differences between our approach and curriculum contrastive learning:
> >
> > A) **Task Focus (Regression vs. Classification):** Curriculum contrastive learning methods, such as CuCo (Chu et al., 2021) and Weatherdepth (Wang et al., 2024), focus on classification or depth estimation tasks, where the difficulty of the tasks is controlled by progressively increasing the complexity of positive or negative sample pairs.
> >
> > In contrast, our approach is designed for continuous regression tasks, such as brain age prediction. The objective here is to accurately map high-dimensional brain imaging data to a continuous age variable, where there is no discrete notion of "easy" or "difficult" classes. Instead, our method focuses on dynamically adjusting the repulsion neighborhood to handle continuous variability in the data, improving model generalization on age-related tasks.
> >
> > B) **Continuous Label Space:** Curriculum learning strategies are typically not designed to handle the continuous label space inherent in regression problems. The contrastive pairs in these methods are organized around discrete labels or tasks that can be ranked in terms of difficulty.
> >
> > Our method is specifically designed to operate in a continuous label space, dynamically adjusting the influence of samples in the embedding space based on their relative proximity to other samples in terms of the continuous age variable.
> >
> >
> >
> > To sum up, we appreciate your comment and would like to clarify that our task, regression, is a well-established area in machine learning with a long history of research and application. Unlike curriculum contrastive learning, which is typically applied to classification tasks with discrete labels, regression tasks have been central to various fields, particularly in predicting continuous outcomes like age, temperature, or financial trends. Our work builds on this long-standing tradition by focusing on continuous variable prediction. In doing so, we introduce Dynamic Localized Repulsion, a novel method designed specifically to enhance model performance for continuous regression tasks like brain age prediction, where capturing subtle, progressive differences is critical.
> >
> > To address this comment, we have updated the manuscript. Particularly, at the end of Section 3, we detail these distinctions more clearly in the revised manuscript to avoid any potential confusion.

---

> > > ### Author Response · Authors · 2024-09-18
> > >
> > > ➡️ **[T#1] Please discuss data size
> > >     - A total of 311 subjects is very limited. This aspect of the study seems critical when comparing different ML approaches.
> > >     - This is also relevant in the choice of the ResNet model pre-trained on OpenBHB (5000 MRI scans from multiple sites).**
> > >
> > > Thank you for your comment. We have added a discussion of the dataset size and its implications in Section 4.1. While the dataset contains only 311 subjects, it represents one of the largest collections of brain stiffness maps, with a wide age range and balanced gender distribution. Additionally, to match the scale of our dataset and avoid overfitting, we selected ResNet-18, the smallest variant of the ResNet architecture, as detailed in Section 4.2. The pre-training on OpenBHB enables the model to leverage learned representations from a larger dataset, which helps mitigate the limitations of our sample size.
> > >
> > >
> > > ➡️ **[T#2] Since data non-uniformity was highlighted as a key challenge, it would help to discuss this more directly.
> > >     - I don't see where Figure 6 is referenced in the paper.
> > >     - Only Figure 3 was discussed, which seemed to suggest that non-uniformity mainly manifests due to the ages of participating subjects.**
> > >
> > > We appreciate the reviewer's feedback. Non-uniformity in our dataset is primarily manifested in the age distribution, as seen in the bimodal distribution shown in Figure 3. Additionally, Figure 6 (now Figure 7) in the appendix highlights the variability in brain coverage across the stiffness maps, which introduces another challenge inherent to this data modality. We have updated the manuscript to ensure both forms of non-uniformity are discussed in Section 4.1.
> > >
> > >
> > > ➡️ **Page-2, top: by focusing on the fine distinctions between similar cases
> > >         - Is this really what the adopted formulation does?
> > >         - [T#3] Please highlight the application of the loss within the embedding space.**
> > >
> > > We agree with the reviewer that the original phrasing could be more precise. We have revised the text to better reflect the mechanism of the contrastive learning approach. The updated manuscript now states that contrastive learning enhances model performance by structuring the embedding space to bring samples with similar ages closer together and push dissimilar samples further apart. This highlights the role of the dynamic localized repulsion loss in structuring the embedding space effectively, focusing on capturing continuous age-related patterns, as explained in Section 3.2.
> > >
> > >
> > > ➡️ **In reference to SimCLR, it is notable that a key finding was the benefit of an additional non-linear projection mapping before computing the distance.
> > >                 - [T#4] It would help to consider this, e.g., as an ablation, or simply mention it for the benefit of follow up work.**
> > >
> > > Thank you for the suggestion. We have conducted an ablation study to evaluate the impact of the non-linear projection mapping, as suggested. Specifically, we compared the model's performance when features are extracted before the projection step (features w/o projection) versus after the projection step (features w/ projection). The results show that the model benefits when features are extracted in the same space as where the distance is computed, achieving better performance with a lower MAE, and this has been added to the manuscript.
> > >
> > > ➡️ **In light of SimCLR and Siamese representations, it is notable that the proposed approach does not include augmentations.
> > >             - [T#5] It would help to consider data augmentation, e.g., as an ablation, or simply mention it for the benefit of follow up work.**
> > >
> > > Thank you for the suggestion. We have conducted an ablation study to evaluate the impact of different data augmentations on the model's performance. Specifically, we applied Noise, Cutout, Rotation, and Flip augmentations and included the results in the manuscript. This study provides insights into how augmentations affect the learned representations in the context of brain imaging data.
> > >
> > >
> > >
> > > ➡️ **Presentation Comments & Nitpicking**
> > >
> > > Thank you for providing a detailed list of presentation-related comments. We have carefully reviewed each point and updated the manuscript accordingly to improve clarity, structure, and overall readability. We have also explored an alternative sampling strategy by introducing a distance-based weighting scheme for non-nearest samples and included this in the appendix. We believe these revisions have enhanced the presentation of our work.

---

> ### Comment · Reviewer_AVKh · 2024-09-20
> **Final Response**
>
> I appreciate the authors' response and revisions, going above and beyond to include several additional ablations. (Note that I only recommended considering those ablations, or simply highlighting the alternative design decisions for future work.)
>
> I also appreciate the authors further elaborating on how they employ contrastive learning for the purposes of regression.  I have revised my review to acknowledge the novelty of this contribution, as evidenced by the improved results for brain age prediction.
>
> That said, I actually don't fully agree with the authors account on a couple of points:
> - Authors assert that existing contrastive learning techniques are designed for classification.  My understanding is that contrastive learning is a general technique for self-supervised learning, where the learned representation can be used for any number of downstream tasks.  Specifically, contrastive learning offers the general recipe which unambiguously appears in all losses in Table 1, as well as the proposed Equations (2) and (3).
> - Similarly, authors assert that curriculum contrastive learning is focused on the relative *difficulty* of examples.  My understanding is that the curriculum is a general recipe for adaptive sampling of examples as training progresses.
> - Authors seem to regard depth estimation as a classification problem, whereas it is routinely approached as a regression problem.
>
> While I appreciate the authors' clarification on the precise scope of the contribution, I'm very much unmoved by the authors' arguments to the contrary of the above.  I am only pushing this point here to help make the contribution easier to appreciate by a wider audience in the ML community.  In more words: I strongly recommend against creating non-essential distinctions between variations of the same algorithm, since I don't view this as progress in the field.  Rather, highlighting commonalities is more likely to help further develop key concepts and facilitate wide adoption.  To sum it up, I invite the authors to reconsider my recommendation for a revised title per **HL#3**.
>
> Finally, a couple comments on the revised draft:
> - I appreciate the additional ablation on the non-linear projection step following SimCLR.  The writing, however, is rather confusing, i.e., last blue paragraph on Page 9. Please revise for clarity. It would help greatly to include the mathematical expression for where the projection is applied.
> - Related to the above, I had requested to provide the mathematical expression for $s_{i, k}$.  I'm guessing this would be $s_{i, k} = d(f(x_i), f(x_k))$, so the projection would then replace this with $d(\pi(f(x_i)), \pi(f(x_k)))$ in Equation (2). (There's really no need to repeat the long expression in Equation (3) again.)

---

> > ### Author Response · Authors · 2024-09-20
> >
> > ➡️ **Positioning & Terminology: I actually don't fully agree with the authors account on a couple of points. To sum it up, I invite the authors to reconsider my recommendation for a revised title per HL#3.**
> >
> > Thank you for your  comments, which have helped us to refine and improve the manuscript. We agree that existing contrastive learning techniques can indeed be used for various downstream tasks, including regression. As such, we have added this to our benchmark as a baseline during the first revision, which showed that it underperforms compared to contrastive losses specifically designed for regression tasks. Additionaly, in response to your feedback on our revisions, we have made the following changes to better align with your suggestions:
> >
> > A) We have added a new discussion focused specifically on curriculum learning, emphasising its general applicability to adaptive sampling of examples during training. This discussion can be found at the end of Section 2, highlighted in blue, and includes the references you suggested.
> >
> > B) As a compromise and to make the contribution more accessible to a wider audience in the ML community as you kindly suggested, we have updated the title to: "Contrastive Regression Learning with Adaptive Neighborhoods for Brain Age Prediction on 3D Stiffness Maps." This revised title reflects the core contribution while avoiding overly specific distinctions. In line with this, we have also removed the paragraph at the end of Section 3 that detailed the disitnction between curriculum contrastive learning and our approach, which had been added during the first revision.
> >
> > ➡️ **Comments for the revised version. I appreciate the additional ablation on the non-linear projection step following SimCLR. The writing, however, is rather confusing, i.e., last blue paragraph on Page 9. Please revise for clarity. It would help greatly to include the mathematical expression for where the projection is applied.
> > Related to the above, I had requested to provide the mathematical expression for. I believe the projection will then replace those with in Equation (2). (There's really no need to repeat the long expression in Equation (3) again.)**
> >
> > Thank you to the reviewer for the constructive feedback.
> >
> > We have made the requested revisions, including clarifying the writing in the last blue paragraph on Page 9 for better readability. Additionally, we have included the mathematical definitions for the projection layer in the ablation study and explicitly defined the similarity score to improve clarity.
> >
> > We opted not to add the projection directly to the loss terms in  (2). Instead, we defined the projection in a way that maintains consistency with the original notation in the loss function to avoid redundancy and ensure the notation remains clear throughout.
> >
> > We appreciate your input and believe these changes enhance the overall clarity  of the manuscript.

---

> ### Comment · Reviewer_AVKh · 2024-09-21
> **Follow up**
>
> I appreciate the authors' engagement and openness to consider my suggestions. It is ultimately up to the authors as to how they choose to present their work.
>
> Regarding the revised title, I prefer to keep the keyword "contrastive learning" intact. There are several examples of modifiers in the literature including (curriculum/decoupled/semi-supervised) contrastive learning. I see the authors' emphasis is on its application to regression problems. However, to me this is taken care of by "brain age prediction." If it helps, this could be rephrased as "brain age regression." Another place where this distinction can be elaborated on more clearly is the abstract and also the introduction. As it stands, attempting to fit this message into the title only makes it less relatable IMHO.
>
> Please also remember to update the title of the submission on OpenReview, once the final version is in.

---

> > ### Author Response · Authors · 2024-09-23
> >
> > ➡️ **Regarding the revised title, I prefer to keep the keyword "contrastive learning" intact. Please also remember to update the title of the submission on OpenReview, once the final version is in.**
> >
> > Thank you for your thoughtful feedback. We have revised the title as suggested and kept the term 'contrastive learning' intact. We also appreciate the reminder to update the title on OpenReview with the final version."

---

### Review · Reviewer_TVUU · 2024-08-27

**Summary Of Contributions:**

The paper proposes a k-nearest neighbors (KNN) based contrastive loss that decreases the number of repulsed samples during training. The paper is the first application of self-supervised learning to brain mechanical properties. The paper shows improved mean absolute error (MAE) for their proposed method compared to a few other contrastive regression losses but does not demonstrate the importance of decreasing the number of repulsed samples during training.

**Audience:**

Yes

**Claims And Evidence:**

Yes

**Requested Changes:**

* Consider adding NNCLR (Dwibedi et al., 2021) as a baseline.
* Consider shrinking Figure 3. I do not think a plot of the distribution of ages should take up half a page.

Requested Clarifications:
* Why are MAE results bolded when intervals overlap? Are these statistically significant results?

**Strengths And Weaknesses:**

Strengths:
* The paper is the first application of self-supervised learning to brain mechanical properties.

Weaknesses:
* The paper does not demonstrate the importance of decreasing the number of repulsed samples during training. It is my understanding that this would recover a weighted Nearest-Neighbor Contrastive Learning of visual Representations (NNCLR) (Dwibedi et al., 2021).
* The method contributions of this paper are not significant. The paper would be a better fit for a healthcare-focused conference or journal.
* The search space for hyperparameters (e.g., initial learning rate and weight decay) is fixed for all methods. There is no detail for hyperparameter tuning for baselines while $NN_{nb,final}$ and $NN_{stepsize}$ are tuned via random search. Are baselines and the proposed method getting the same time and compute resources for hyperparameter tuning?

Debidatta Dwibedi, Yusuf Aytar, Jonathan Tompson, Pierre Sermanet, Andrew Zisserman. With a Little Help From My Friends: Nearest-Neighbor Contrastive Learning of Visual Representations. In *Proceedings of the IEEE/CVF International Conference on Computer Vision (ICCV)*, 2021.

---

> ### Author Response · Authors · 2024-09-18
>
> ➡️ **The paper does not demonstrate the importance of decreasing the number of repulsed samples during training. It is my understanding that this would recover a weighted Nearest-Neighbor Contrastive Learning of visual Representations (NNCLR) (Dwibedi et al., 2021).**
>
> We thank the reviewer for the insightful comment. We acknowledge that the proposed dynamic localized repulsion may resemble a weighted form of NNCLR (Dwibedi et al., 2021) in some respects, particularly regarding the use of neighbors. However, our method diverges significantly from NNCLR, particularly in its applicability to regression tasks and its dynamic adjustment of repulsed samples during training.
>
> **Regression vs. Classification:** While NNCLR was designed for classification tasks, our method is tailored to regression, specifically brain age prediction. In regression, the continuous nature of the target variable (age) requires a fundamentally different approach to learning similarities between samples. The use of dynamic weighting based on the similarity in age, as well as dynamically adjusting the number of repulsed samples, is critical for handling this continuous prediction problem.
>
> **Dynamic Adjustment of Neighbors:** A core innovation in our method is the progressive reduction in the number of repulsed samples throughout the training process. In contrast, NNCLR operates with a fixed set of nearest neighbors throughout training. The dynamic nature of our approach allows for a broad generalization early in training, followed by a more focused refinement on the localized relationships between samples as the model converges. This is crucial for accurately predicting brain age, where finer distinctions between samples are needed over time.
>
> To address this comment, we added a clarifying note at the beggining of subsection 3.1. The changes can be found in blue colour.
>
>
>
> ➡️ **The method contributions of this paper are not significant. The paper would be a better fit for a healthcare-focused conference or journal.**
>
> We appreciate the reviewer's feedback. However, we would like to clarify that TMLR explicitly welcomes research that applies machine learning across various domains, including healthcare. Numerous papers focused on healthcare applications have been published in TMLR, demonstrating that this journal is an appropriate venue for impactful interdisciplinary work.
>
> Our paper, which proposes a novel machine learning method tailored to brain age prediction, addresses critical challenges in the healthcare domain, while also introducing advancements in contrastive learning and regression tasks that are relevant to the broader machine learning community. We believe this makes our work a strong fit for TMLR.
>
>
> ➡️ **The search space for hyperparameters (e.g., initial learning rate and weight decay) is fixed for all methods. There is no detail for hyperparameter tuning for baselines while  and  are tuned via random search. Are baselines and the proposed method getting the same time and compute resources for hyperparameter tuning?**
>
> Thank you for your question regarding the fairness of hyperparameter tuning across methods. We can confirm that all experiments were run under fair conditions. For all experiments, we used the same fixed hyperparameters (e.g., initial learning rate and weight decay) to ensure a fair comparison. The only difference is that our proposed method introduces two additional parameters, which were the only ones subjected to tuning. These additional parameters were tuned via random search, but all other hyperparameters remained consistent with the baselines to ensure a fair allocation of time and compute resources.

---

> > ### Author Response · Authors · 2024-09-18
> >
> > ### Requested Changes:
> > ➡️ **Consider adding NNCLR (Dwibedi et al., 2021) as a baseline**
> >
> > Thank you for the suggestion. While we clarify that NNCLR (Dwibedi et al., 2021) is designed for classification tasks and not for regression, we acknowledge the relevance of evaluating similar methods. In response, we have conducted two additional experiments using SimCLR (Chen et al., 2020a) and NNCLR (Dwibedi et al., 2021). These results, which can be found in Table 3, further demonstrate the performance gap due to the differences in task nature and the advantages of our carefully designed model for brain age regression.
> >
> >
> > ➡️ **Consider shrinking Figure 3. I do not think a plot of the distribution of ages should take up half a page.**
> >
> > Thank you for the suggestion. We have resized Figure 3 to make better use of the space, ensuring that it no longer occupies half a page.
> >
> >
> >
> > ➡️ **Why are MAE results bolded when intervals overlap? Are these statistically significant results?**
> >
> > Thank you for your observation regarding the bolding of results. The bolding in Table 3 is intended to highlight the best-performing value, not necessarily statistical significance. We would like to emphasise that the confidence intervals reflect variability across the different test sets and random seeds used in the experiments. While the overlap indicates that the performance differences might not be significant at a conventional threshold (e.g., p < 0.05), the mean values consistently favor our proposed method.
> >
> > We also clarify that the confidence intervals are based on a certain number of trials, and despite the overlap, our method shows consistent improvement across experiments.
> >
> > To avoid any potential confusion regarding the bold results, we now highlight the best performance in green.

---

> ### Comment · Reviewer_TVUU · 2024-09-21
> **Official Comment by Reviewer TVUU**
>
> Thank you for your detailed response. I have updated the **Audience** section of my review.
>
> Could you please clarify how SimCLR and NNCLR are being applied to your regression problem? My intention in mentioning NNCLR, was that without a baseline or ablation of your proposed method that does not decrease the number of repulsed samples during training the paper lacks evidence that this adjustment is critical.

---

> > ### Author Response · Authors · 2024-09-23
> >
> > ➡️ **Thank you for your detailed response. I have updated the Audience section of my review. Could you please clarify how SimCLR and NNCLR are being applied to your regression problem?**
> >
> > Thank you for your question. We acknowledge that data augmentations are required for contrastive learning frameworks like SimCLR and NNCLR, and in our case, Gaussian Noise was applied. This clarification has been added to the "Evaluation Protocol" section in the manuscript. In terms of their application to our regression problem, analogously to the other compared methods, and as mentioned in the Evaluation Protocol, the pre-trained ResNet-18 underwent full fine-tuning using the respective SimCLR and NNCLR losses on our brain stiffness dataset to learn representations. Following the training of the representations, we employed a Ridge Regression estimator on top of the frozen encoder to predict age.
> >
> > ➡️ **My intention in mentioning NNCLR, was that without a baseline or ablation of your proposed method that does not decrease the number of repulsed samples during training the paper lacks evidence that this adjustment is critical.**
> >
> > Thank you for your insightful comment. In addition to adding NNCLR to the benchmark, which indeed uses nearest neighbors without decreasing the set of nearest neighbors during training, we would like to clarify that the requested baseline where our proposed method does not decrease the number of repulsed samples is already included in the manuscript. This is represented by the "Exponential" method in our benchmark. The Exponential method does not reduce the set of nearest neighbors throughout training, while also leveraging continuous label information to tailor it specifically for regression tasks. This provides a direct comparison between a method that keeps the repulsion set static (Exponential) and our dynamic localized approach.

---

> ### Comment · Reviewer_TVUU · 2024-10-01
> **Official Comment by Reviewer TVUU**
>
> Thank you for this clarification. I have updated my review. I suggest highlighting this in the paper for those unfamiliar with the Exponential method.

---

> > ### Author Response · Authors · 2024-10-07
> >
> > Thanks for all your feedback to improve our work.

---

### Review · Reviewer_58LT · 2024-09-05

**Summary Of Contributions:**

This paper presents a novel approach for brain age prediction using contrastive regression techniques applied to brain stiffness maps. The proposed method, Dynamic Localized Repulsion, addresses the challenge of non-uniform data distributions in medical imaging, where the method gradually drops negative samples that are farther away from the anchor when calculating the contrastive objective. By focusing on localized neighborhoods during training, it reduces the effect of oversampled instance classes on the learning. The new contrastive objective is shown to improve the predictive power of models working with mechanical brain properties, which are highly sensitive to age-related changes.

**Audience:**

Yes

**Claims And Evidence:**

Yes

**Requested Changes:**

See Weaknesses Above.

**Strengths And Weaknesses:**

Strengths:
1. Few researches have attempted to improve contrastive learning by focusing on negative samples in loss functions. The proposed method pioneers negative sample selection in contrastive regression problems, which is intriguing.
2. The intuition behind the method is well-aligned with the existing challenge of non-uniform data distributions in medical imaging.
3. The formulation of the objective function is straightforward, and the idea is easy to follow.

Weaknesses:
1. Despite the fact that majority class instances may negatively affect the representation learning for minority class instances, directly dropping them seems to impact the generalization of the learned representations. For example, it has been shown that some hard negative instances are beneficial to contrastive learning [1]. Considering this, I believe reweighting the negative samples based on distance might be more effective than simply dropping them during training.
2. The framework builds upon Contrastive Regression Loss (CRL) [2], with modifications in the sampling of negative pairs. In the experiments from [2], it was observed that CRL alone might not be sufficient to learn good representations, thus requiring auxiliary losses (such as L1) to support the learning process. I wonder if this is also the case for the objective function introduced in this paper.

[1] Robinson, J., Chuang, C.Y., Sra, S. and Jegelka, S., 2021, January. CONTRASTIVE LEARNING WITH HARD NEGATIVE SAMPLES. In International Conference on Learning Representations (ICLR).

[2] Zha, K., Cao, P., Son, J., Yang, Y. and Katabi, D., 2024. Rank-n-contrast: learning continuous representations for regression. Advances in Neural Information Processing Systems, 36.

---

> ### Author Response · Authors · 2024-09-18
>
> ➡️ **Despite the fact that majority class instances may negatively affect the representation learning for minority class instances, directly dropping them seems to impact the generalization of the learned representations. For example, it has been shown that some hard negative instances are beneficial to contrastive learning [1]. Considering this, I believe reweighting the negative samples based on distance might be more effective than simply dropping them during training.**
>
>
> Thank you for highlighting the potential drawbacks of directly dropping majority class instances, particularly in relation to contrastive learning. In line with your suggestion, we have explored an alternative strategy by introducing a distance-based weighting scheme for non-nearest samples. We conducted an ablation study (now included in the manuscript appendix) to evaluate the effect of different weighting parameters on model performance.
>
> Our study found that, while the distance-based reweighting strategy allows us to retain useful negative samples, only one model using an intermediate weighting parameter performed slightly better compared to the baseline where non-nearest samples were simply dropped. This suggests that reweighting provides some benefit in specific cases, though its overall impact on performance was modest.
>
> As this is a first strategy to leverage non-nearest samples, we acknowledge that more advanced methods to introduce bias in the sampling strategy could be explored. This direction, including adaptive or more dynamic reweighting mechanisms, is an exciting area for future work and beyond the scope of this study.
>
> To sum up, we included new experiments on different weighting parameters—these can be seen in page #9. Moreover, we added in the conclusion, as future work, the potential of advanced methods to improve sampling, which could lead to more significant performance improvements across various contexts.
>
>
>
> ➡️ **The framework builds upon Contrastive Regression Loss (CRL) [2], with modifications in the sampling of negative pairs. In the experiments from [2], it was observed that CRL alone might not be sufficient to learn good representations, thus requiring auxiliary losses (such as L1) to support the learning process. I wonder if this is also the case for the objective function introduced in this paper.**
>
>
> Thank you for your comment. In response, we conducted a similar experiment to the one referenced in Zha et al. (2024) to investigate whether auxiliary losses (such as MSE, L1, Huber, and DEX) could improve the performance of our proposed framework. The findings have been appended to the manuscript. The new experiments can be found in Figure 4b.

---

### Decision · Action_Editor_re3p · 2024-10-25

**Recommendation:** Accept with minor revision

**Comment:**

The paper introduces a novel supervised contrastive learning method in the context of brain imaging. All reviewers voted for accepting the paper.

The reviewers appreciated the novelty and strong empirical results in predicting the age based on brain imagining. The experiments were done on a convincingly broad set of datasets. Predicting the age based on brain stiffness maps was found novel and interesting.

The reviewers were not convinced about the impact of the proposed contrastive method beyond the scope of the work. I am leaning to agree with the reviewers. The method was not compared to other natural methods (e.g. purely supervised learning) nor tested on datasets from other domains. Hence, it is not straightforward whether the method will work in a broader context. Having said that, the proposed method is interesting and natural, hence it indeed might be of an interest to the contrastive learning community.

All in all, it is my pleasure to recommend accepting the paper. Please make sure to address comments made by reviewers. I would suggest adding a bit more clarity in writing, the method was not tested on enough different datasets to fully claim it is superior to other approaches. It would be fair to say it is a promising method.

**Audience:**

The reviewers were convinced that the results will be interested for the brain imagining community. Some of the reviewers voiced the opinion that the contrastive learning method might be less appealing to the broader machine learning community. I think it will be likely of interest to some, but many might find the experiments not broad enough to prove yet that the proposed method would help on different datasets.

**Claims And Evidence:**

The reviewers appreciated the experiments and the key claims about the performance on the age prediction task are well supported.

The paper also makes claims regarding the efficacy of the proposed supervised contrastive learning, which I feel are less well supported because the method was not tested beyond the domain of brain imagining, nor was it compared to other approaches (e.g. pure supervised learning).  I would suggest adding a bit more clarity in writing, the method was not tested on enough different datasets to fully claim it is superior to other approaches. It would be fair to say it is a promising method.